# Apparent Diffusion Coefficient Metrics to Differentiate between Treatment-Related Abnormalities and Tumor Progression in Post-Treatment Glioblastoma Patients: A Retrospective Study

**DOI:** 10.3390/cancers15204990

**Published:** 2023-10-14

**Authors:** Rik van den Elshout, Siem D. A. Herings, Manoj Mannil, Anja M. M. Gijtenbeek, Mark ter Laan, Robert J. Smeenk, Frederick J. A. Meijer, Tom W. J. Scheenen, Dylan J. H. A. Henssen

**Affiliations:** 1Department of Medical Imaging, Radboud University Medical Center, 6525 GA Nijmegen, The Netherlands; siem.herings@radboudumc.nl (S.D.A.H.); anton.meijer@radboudumc.nl (F.J.A.M.); tom.scheenen@radboudumc.nl (T.W.J.S.); dylan.henssen@radboudumc.nl (D.J.H.A.H.); 2Radiologie Radboudumc, Geert Grooteplein Zuid 10, 6525 GA Nijmegen, The Netherlands; 3University Clinic for Radiology, Westfälische Wilhelms-University Muenster and University Hospital Muenster, Albert-Schweitzer-Campus 1, DE-48149 Muenster, Germany; manoj.mannil@ukmuenster.de; 4Department of Neurology, Radboud University Medical Center, 6525 GA Nijmegen, The Netherlands; anja.gijtenbeek@radboudumc.nl; 5Department of Neurosurgery, Radboud University Medical Center, 6525 GA Nijmegen, The Netherlands; mark.terlaan@radboudumc.nl; 6Department of Radiation Oncology, Radboud University Medical Center, 6525 GA Nijmegen, The Netherlands; robertjan.smeenk@radboudumc.nl

**Keywords:** apparent diffusion coefficient, diffusion imaging, glioblastoma, treatment-related abnormality, tumor progression, receiver operating characteristic curve

## Abstract

**Simple Summary:**

Patients suffering from glioblastoma receive treatment according to the Stupp protocol. After chemoradiotherapy, the glioblastoma either continue to develop, or treatment-related vascular damage comes into play, both causing new, seemingly identical contrast-enhancing lesions in follow-up MRI, where clinicians and radiologists alike can only differentiate between tumor progression (TP) and treatment-related abnormalities (TRA) by seeing the temporal evolution of the lesion and the patients clinical status. We investigate the use of diffusion MRI and the quantitative apparent diffusion coefficient (ADC) in a retrospective patient cohort and we sought to validate a previously reported ADC cutoff value for its distinctive properties between TP and TRA. In the end, ADC values were shown not to be highly discriminative and, compared to the mean ADC values between TP and TRA previously reported in the literature, are more overlapping than expected.

**Abstract:**

Distinguishing treatment-related abnormalities (TRA) from tumor progression (TP) in glioblastoma patients is a diagnostic imaging challenge due to the identical morphology of conventional MR imaging sequences. Diffusion-weighted imaging (DWI) and its derived images of the apparent diffusion coefficient (ADC) have been suggested as diagnostic tools for this problem. The aim of this study is to determine the diagnostic accuracy of different cut-off values of the ADC to differentiate between TP and TRA. In total, 76 post-treatment glioblastoma patients with new contrast-enhancing lesions were selected. Lesions were segmented using a T1-weighted, contrast-enhanced scan. The mean ADC values of the segmentations were compared between TRA and TP groups. Diagnostic accuracy was compared by use of the area under the curve (AUC) and the derived sensitivity and specificity values from cutoff points. Although ADC values in TP (mean = 1.32 × 10^−3^ mm^2^/s; SD = 0.31 × 10^−3^ mm^2^/s) were significantly different compared to TRA (mean = 1.53 × 10^−3^ mm^2^/s; SD = 0.28 × 10^−3^ mm^2^/s) (*p* = 0.003), considerable overlap in their distributions exists. The AUC of ADC values to distinguish TP from TRA was 0.71, with a sensitivity and specificity of 65% and 70%, respectively, at an ADC value of 1.47 × 10^−3^ mm^2^/s. These findings therefore indicate that ADC maps should not be used in discerning between TP and TRA at a certain timepoint without information on temporal evolution.

## 1. Introduction

Radiological follow-up of glioblastoma is complicated by the fact that chemoradiation therapy can lead to a new contrast-enhancing lesion that does not reflect tumor progression (TP). Contrary to TP, these lesions regress over time and are known as treatment-related abnormalities (TRA). In the literature, TRAs are subdivided into two lesion types: pseudoprogression and radiation necrosis, although these terms seem to be used interchangeably. To differentiate pseudoprogression and radiation necrosis from TP, temporal evolution is essential. Pseudoprogression is defined as radiographic evidence resembling disease progression, typically within 3–6 months post-treatment, often followed by spontaneous involution or clinical improvement without additional treatment [1], whereas radiation necrosis often occurs 9–12 months after treatment, though the radiological effects can be sustained in a stable state for months [2]. TP, on the other hand, shows growth over time when no action is taken, leading to a worse prognosis with a greatly reduced overall survival [3].

The methylation status of the O^6^-methylguanine-DNA methyltransferase (MGMT) promoter is an important biomarker and predictor for treatment reaction to alkylating agents such as temozolomide. The gene is responsible for a protein involved in DNA repair. Methylation of the promoter of this gene reduces tumor capability for DNA repair after temozolomide therapy, leading to increased tumor cell death. TRA is more likely to occur in tumors with hypermethylation of the MGMT promoter, indicating a more favorable outcome [4,5].

Currently, the gold standard to distinguish TRA from TP consists of histopathological examination of brain tissue acquired through biopsy or resection [6]. Non-invasively, TRA and TP can be distinguished by the longitudinal use of various imaging techniques, including multiparametric MRI to evaluate the temporal evolution [7]. However, serial imaging of the brain with an interval of several months leads to a delay in treatment in the case of TP. Therefore, the development of robust imaging biomarkers for early detection of TP on a single MRI session is much needed.

Shortly after treatment, TP and TRA have an identical appearance on single-timepoint conventional magnetic resonance imaging (MRI) sequences (T1, post-contrast T1, T2, FLAIR). Previous studies showed that the use of these sequences alone cannot adequately distinguish TRA from TP [8,9,10]. Recent small-sampled studies suggested that diffusion-weighted imaging (DWI) could be a possible, non-invasive alternative for early detection of TP [11,12,13]. The use of the derived apparent diffusion coefficient (ADC) values has especially been suggested as a promising quantitative MRI biomarker. ADC values are a measure of the apparent magnitude of diffusion of water in tissue and can be considered a surrogate marker of cellular density [14], as cellular compartmental restrictions hinder the free motion of water molecules. The ability of the ADC to differentiate between TP and TRA can be attributed to this, where cellular density is increased in TP.

A recent meta-analysis from van den Elshout et al. on the use of ADC values as an imaging biomarker to differentiate TP from TRA showed promising results, with a pooled sensitivity of 85% (95% CI 78.5–89.8%) and pooled specificity of 81% (95% CI 72.3–86.6%) [15]. However, a lack of harmonization of imaging protocols and a wide range of cut-off values throughout the literature severely hamper further implementation. In addition, as is common in this field of research, diagnostic accuracy and optimal cut-off values were all based on one cohort without the use of an external validation cohort, and the predictive value of such a cut-off value needs testing on new patient cohorts.

The aim of this paper is to determine whether ADC values could be useful in distinguishing between TP and TRA in post-treatment glioblastoma patients. First, a large cohort of patients with new contrast-enhancing lesions, developed after primary surgery and adjuvant treatment, were evaluated by use of the ADC values alone. The diagnostic accuracy of the ADC values was compared with the diagnostic accuracy of multiple radiologists’ readings of the multi-parametric MRI data, including dynamic susceptibility in contrast to perfusion-weighted MRI, and both were compared to gold standard diagnosis by follow-up according to the Response Assessment in Neuro-Oncology (RANO) criteria [16]. Second, the pooled cut-off value of ADC to distinguish TP from TRA, which was derived from the aforementioned meta-analysis, was tested in this cohort as external validation.

## 2. Materials and Methods

Anonymous imaging data of post-treatment histopathologically proven glioblastoma patients were included retrospectively. The Picture Archiving and Communication System (PACS) of our University Medical Center was searched by a clinical investigator (D.H.) between 2019 and 2022. The patient population is shown in Table 1. Patients were eligible for this study when (i) they were treated according to current guidelines with tumor resection, followed by chemoradiation therapy, according to the Stupp protocol [17], (ii) a new contrast-enhancing lesion was observed on follow-up imaging, (iii) when ADC maps were available and (iv) when clinical and radiological follow-up for at least 3 months provided a final diagnosis of the entity of the new contrast-enhancing lesion. Diagnosis of either TP or TRA was made according to the RANO criteria [16]. In short, the RANO criteria categorize tumor response into four main categories based on imaging features and clinical assessment, and comply with the subsequent requirements: complete response shows the disappearance of all enhancing lesions sustained for four weeks, no new lesions and clinical stability without the requirement of high dose corticosteroids; partial response also shows no new lesions, a 50% or more decrease in measurable enhancing lesions for four weeks and patients show clinical improvement or stability with reduced corticosteroid use compared to baseline; stable disease does not qualify for complete or partial response, but with stable, non-enhancing T2 lesions and clinically improved or stable patients with consistent or reduced corticosteroid dosage requirement; progression shows an increase in contrast-enhancing lesions, T2 lesions and new lesions with clinical deterioration which is likely tumor-related. Although histopathological assessment of the new contrast-enhancing lesion is described in the literature as the gold standard diagnosis, the Dutch guidelines [18] state that in the follow-up of glioblastoma patients, multiparametric radiological follow-up should suffice in the differentiation between TP and TRA, and reoperation with histopathology is a strategy reserved for cases in which a clear diagnosis cannot be made. Exclusion criteria comprised: (1) lesions smaller than 3 mm, (2) infratentorial lesions, (3) non-contrast-enhancing lesions and (4) patients in whom no consensus diagnosis was reached. In addition, patients were excluded when they did not choose to share their data for scientific purposes. Ethical approval was waived due to the retrospective nature of the study.

### 2.1. MRI Acquisition Protocol

All MR imaging sessions were performed on 1.5-Tesla Siemens MAGNETOM Avanto 1.5 T MRI scanners (Siemens Medical Solutions, Erlangen, Germany). The MRI protocol is included in Table 2. The scanning protocol included a T1-weighted MPRAGE sequence and a T1-weighted SPACE sequence (TR 600 ms; TE 7.1 ms; slice thickness 1 mm) before and after the application of gadoteric acid (Dotarem, Guerbet, France) as a contrast agent. The exact amount of contrast agent administered differed per individual, with 0.1 mg/kg as a preloaded dose at an infusion speed of 5 mL/s. The protocol also contained an axial T2-weighted sequence, a FLAIR and a b= 0 s/mm^2^ and b = 1000 s/mm^2^ DWI sequence. Thereafter, a second injection of Dotarem 20 mL was given to perform DSC perfusion-weighted MRI.

### 2.2. Obtaining and Analyzing the ADC Values

The ADC values were obtained by analyzing the new contrast-enhancing lesions on the post-contrast T1-weighted MRI. The regions of interest (ROI) consisting of contrast-enhancement were semi-automatically segmented in ITK-SNAP (version 3.8.0) [19], not taking into account any areas outside or within the contrast-enhancing regions. ITK-SNAP is a software application used to segment structures in 3D medical images, using active contour methods, manual delineation and image navigation. If deemed necessary, ROIs were adapted by the use of manual segmentation. All segmentations were carried out by two researchers (R.E. and S.H.) and were reviewed by an experienced radiology resident with 7 years of experience and a main focus in experimental neuroradiology (D.H.). The ADC maps were then linearly resliced and transformed to match the voxel grid of the T1 sequences using the registration tool of ITK-SNAP. This allowed for the retrieval of the mean, 25th and 75th percentiles of the ADC values of the ROIs, using the FMRIB Software Library (FSL) (version 6.0.5) FSLstats function of FSL utilities [20].

### 2.3. Radiological Reading of Multiparametric MRI Data

Due to the retrospective and clinical nature of this study, the corresponding radiology reports were available for analysis. All MRI scans had been clinically assessed by board-certified neuro-radiologists with experience ranging from 7 years to 20 years. The multiparametric data that were assessed comprised all acquired scans according to the scanning protocol. The results of radiology reports were independently dichotomized to an outcome of either TRA or TP. The actual outcome was determined over the course of at least three months following the RANO criteria [16]. A confusion matrix was set up comparing the radiological reading to the outcome, leading to a sensitivity and specificity for multiparametric reading.

### 2.4. Testing the Literature Derived Cut-Off Value of the ADC Value to Distinguish TP from TRA

In a recent publication of van den Elshout et al. [15], the diagnostic accuracy of ADC values throughout the literature was analyzed. This resulted in a mean ADC value of TP lesions of 1.13 × 10^−3^ mm^2^/s (95% confidence interval: 0.91 × 10^−3^–1.32 × 10^−3^ mm^2^/s), whereas TRA lesions showed a mean ADC value of 1.38 × 10^−3^ mm^2^/s 95% confidence interval: 1.33 × 10^−3^–1.45 × 10^−3^ mm^2^/s). Based on these values and the 95% confidence intervals, a cut-off ADC value of 1.33 × 10^−3^ mm^2^/s was chosen to calculate sensitivity and specificity. Higher values were assumed to reflect TRA, whereas lower values were assumed to be a region of TP.

### 2.5. Statistical Assessment

Statistical analyses were carried out by use of IBM SPSS Statistics for Windows (version 28.01.1). The normality of data was assessed using the two-sample Kolmogorov–Smirnov test. Depending on the normality of data, the proper statistical test was used to assess differences in ADC values between the TRA and TP groups. Levene’s test was used to determine the homogeneity of variance. A binary logistic regression was used to predict outcomes based on a combination of MGMT status and mean ADC value.

A receiver operating characteristic (ROC) curve was generated to determine a cut-off value with an optimal sensitivity-specificity ratio according to Youden’s J index to be used to classify TRA and TP. The diagnostic accuracy of this curve was determined by the area under the curve (AUC). The outcome of this cut-off value was compared with the literature-based cut-off value. In addition, the sensitivity and specificity of the reading of multiparametric data were determined by comparing the dichotomized reports with the ground truth, which was based on imaging results, multidisciplinary consultation and follow-up. In all analyses, the positive state was defined as TRA. The frequencies of the methylation status of the MGMT promoter were compared between groups of patients with TP and TRA after testing for homogeneity and normality.

## 3. Results

### 3.1. Included Patients

Seventy-six glioblastoma patients met the inclusion criteria. Clinical and radiological follow-up showed 26 patients to have TRA, whereas 50 patients suffered from TP. An overview of the included patients, including the methylation status of the MGMT-promoter gene and isocitrate dehydrogenase (IDH) mutation status can be appreciated in Table 1.

### 3.2. Comparison of ADC Values in TP and TRA

In all patients, we could successfully transpose the ROI segmented on the post-contrast T1w MRI to the ADC maps. An example of the segmentation to find the mean ADC value of TP and an example of TRA can be appreciated in Figure 1. The Kolmogorov–Smirnov test showed that the mean ADC values were normally distributed (D(76) = 0.079; *p* = 0.200). Levene’s test was observed to be non-significant (F = 0.380; *p* = 0.539), indicating equal variances. The mean ADC value found in TRA lesions (mean = 1.53 × 10^−3^ mm^2^/s; SD = 0.28 × 10^−3^ mm^2^/s) compared to TP lesions (mean = 1.32 × 10^−3^ mm^2^/s; SD = 0.31 × 10^−3^ mm^2^/s) was found to differ significantly (t(74) = 2.87; *p* = 0.003 one-tailed; Cohen’s *d* = 0.298).

### 3.3. Methylation Status of the MGMT Promoter

Twenty-four patients with TRA and forty-three patients suffering TP were eligible for analysis. For the remaining nine patients, the methylation status of the MGMT promoter was unknown. Levene’s test for homogeneity of variance indicated heteroscedastic data for MGMT status (*p* < 0.001). Welch’s *t*-test showed that hypermethylation of the MGMT promoter was significantly more common in the group of patients with a TRA lesion (*p* < 0.001). Patients with hypermethylation of the MGMT promoter had a 50% chance for TRA instead of TP, whereas non-hypermethylated MGMT promoter patients only had a 12% chance for TRA, corroborating previous literature stating MGMT promoter methylation status can predict the incidence and outcome of pseudoprogression [21].

### 3.4. Comparison of ADC Values in Lesions with Hypermethylation of the MGMT Promoter vs. Lesions without the Hypermethylation of the MGMT Promoter

Levene’s test for homogeneity of variance showed equal variances between groups (*p* = 0.206). When comparing the mean ADC values between lesions with hypermethylation of the MGMT promoter vs. lesions without the hypermethylation of the MGMT promoter, no significant differences were found (*p* = 0.491).

### 3.5. Diagnostic Accuracy

An ROC analysis of the ADC values (Figure 2) yielded an AUC of 0.71 (95% confidence interval: 0.58–0.83) with regard to distinguishing TP and TRA on mean ADC values alone. A Youden’s index optimal ADC cut-off value of 1.47 × 10^−3^ mm^2^/s yielded a sensitivity and specificity of 65% and 70%, respectively. The 25th and 75th ADC percentiles performed worse with an AUC of 0.60 (95% CI: 0.47–0.73) and 0.65 (95% CI: 0.52–0.78), respectively.

Interestingly, when combining MGMT status and mean ADC value into a predicted probability using binary logistic regression, diagnostic accuracy is increased, with an AUC of 0.78 (95% CI: 0.66–0.90), a sensitivity of 85% and a specificity of 68% at the optimal point according to Youden’s J index. This did not yield an optimal ADC cutoff but a predicted probability of TRA as an outcome.

### 3.6. External Validation on Literature Derived Cut-Off

The literature-based cut-off ADC value of 1.33 × 10^−3^ mm^2^/s yielded a sensitivity and specificity of 77.0% and 50.0% in our patient cohort.

### 3.7. Radiological Reading of Multiparametric MRI Data

The sensitivity and specificity of the diagnosis based on the dichotomized TP/TRA radiology reports of the multiparametric MRI data were assessed for the total patient group, using the temporal evolution according to RANO as the ground truth. This analysis resulted in a sensitivity of 35% and a specificity of 90% with TRA as the positive state, as can be seen in Table 3.

## 4. Discussion

This study confirms that mean ADC values differ between TP and TRA lesions on a group level (*p* = 0.039), where ADC values are significantly higher in TRA than in TP. However, due to the considerable overlap between the groups, ADC values for the individual patient performed with a mediocre diagnostic capacity to distinguish TP from TRA. Radiological readings of multiparametric MRI data, including dynamic susceptibility contrast and perfusion-weighted MRI sequences, showed equal diagnostic accuracy, with a preference for overcalling TP from a clinical point of view, in order to diagnose and treat progressing tumor tissue accordingly. This shows that determining an optimal trade-off between sensitivity and specificity is highly dependent on the clinical context and the treatment goal determined by the treating physician and the patient.

The presented data show that the sole use of quantitative ADC values in the radiological follow-up of post-treatment glioma patients is not feasible and highlight the importance of multiparametric radiological follow-up in these patients. However, the difference in ADC values between TP and TRA lesions corroborates the theory that TP has a low ADC value due to increased cellularity and reduced free water diffusion, whereas TRA lesions have a high ADC value due to the increased free diffusion of water molecules caused by treatment-related damage and possible edema [6].

When comparing diagnostic accuracies of studies on the use of ADC values to distinguish TRA from TP, the reported sensitivity and specificity values (65%/70%, respectively) were remarkably lower than values reported in recent meta-analyses, with reported sensitivity and specificity values of 95/83% [22], 85/81% [15] and 82/84% [23], respectively. The papers in the meta-analyses had relatively small sample sizes, all using different parameters such as mean ADC, relative ADC or fifth percentile ADC, indicating that there is no harmonized way to demonstrate the use of ADC in distinguishing TP from TRA and suggesting that the data collection and analysis methods presented in these papers may be different from our work. Additionally, a publication bias towards positive results, as well as preliminary results in small patient cohorts, may have played a role in the higher values for sensitivity and specificity reported in the mentioned meta-analyses. Moreover, different centers use different MR systems, imaging methods, acquisition protocols and/or segmentation methodologies, leading to differences in ADC values and performances between studies. Altogether, cutoff values in a single center optimized for a single scanning protocol could be useful if verified on a prospective cohort. But these data cannot be extrapolated to different centers.

Next, regarding the use of ADC maps, various MRI sequences have been developed and investigated to allow early detection of TP, including perfusion-weighted imaging and magnetic resonance spectroscopy [24,25,26]. All different sequences reported good to excellent differentiating ability, albeit mostly on a group level in retrospective, relatively small sampled cohort studies, often without the use of validation cohorts [27]. Prospective studies, in particular those using external validation cohorts, are rare, and randomized controlled trials seem unavailable. The paradox arises that advanced imaging is not part of response assessment criteria, which causes the lack of harmonized guidance on its use, while at the same time, the lack of standardization severely draws back the systematization of uniform guidelines. This lack of harmonization inhibits the further development and overall clinical implementation of these techniques for quantitative analysis of imaging metrics. By standardizing imaging protocols—for this study, diffusion imaging protocols specifically—data become more reproducible and the exchange of data between centers according to FAIR principles (findable, accessible, interoperable and reusable) enables validation of external cohorts more readily. In order to harmonize and quantify data, consistent use of the Imaging Biomarker Standardization Initiative (IBSI) and the use of standardized methods and metrics are examples of how to achieve quantification. In order to harmonize diffusion data, we would also suggest using b = 50 s/mm^2^ instead of b = 0 s/mm^2^ as the lowest applied diffusion-encoding gradient to reduce fast signal attenuation due to perfusion effects, which is currently highly likely to confound diffusion data. The new quantified, harmonized data need to be validated in prospective large clinical cohorts in order to reliably draw conclusions on their clinical use.

These data support the current literature, proving that the methylation status of the MGMT promoter is strongly correlated with patient outcome and functions as an important biomarker for clinical outcome. Several clinical trials have demonstrated the prognostic significance of the methylation status of the MGMT promoter in glioblastoma patients [28,29,30]. Patients with a hypermethylated MGMT promoter were found to have longer overall survival and progression-free survival compared to patients without MGMT promoter methylation. Moreover, hypermethylation of the MGMT promoter has been used to stratify patients for clinical trials and treatment decisions. For example, the Stupp protocol, which is the current standard of care for glioblastoma, recommends temozolomide as part of the treatment regimen for patients with MGMT promoter methylation, while patients without MGMT promoter methylation may benefit from alternative treatment options [31].

What stands out is that, in contrast to the multiparametric reading as a clinical example, the combination of MGMT status with the mean ADC value has a remarkably higher sensitivity and specificity. This begs the question of whether there are more insights to gain when combining molecular biomarkers with radiological quantification. Additionally, the 2021 WHO classification of glial tumors introduces new molecular biomarkers for the classification of diffuse gliomas. Markers like Epidermal Growth Factor Receptor (EGFR) amplifications, Telomerase Reverse Transcriptase (TERT) promoter mutations, combined whole chromosome 7 gain, and combined whole chromosome 10 loss are crucial for diagnosing glioblastoma IDH wildtype [32]. Several other molecular determinants are associated with tumor aggressiveness. For instance, Platelet-Derived Growth Factor Receptor Alpha (PDGFRA) expression trends show differences in overall survival among glioblastoma patients [33]. However, none of these biomarkers have yet approached the prognostic significance of MGMT status. This highlights the apparent lack of knowledge and understanding of molecular subclassifications in glioblastoma [34].

### Strengths and Limitations

A particular strength of the current study is that our cohort served as a validation cohort of the previously reported cut-off value in the literature. As previous research suggested that the segmentation of post-treatment glioblastoma lesions can be difficult [35], our semi-automatic multi-reader segmentation methodology accurately covering the entire contrast-enhancing lesion can also be considered a strength of our study protocol. This approach reduces inter- and intra-observer variability and provides an overview of the mean ADC value of the entire contrast-enhancing lesion. Another strength of the current study is the fact that for the first time, ADC values of entire lesions were compared with standard clinical reading of multiparametric data in the post-treatment evaluation of gliomas to distinguish TP from TRA.

However, an important limitation is the retrospective design of this study, lacking prospective validation cohorts, as well as the lack of obtained images at higher B values (3000 s/mm^2^) which would provide better contrast reflecting diffusivity and less T2 shine-through [36]. A weakness is that this method ignores the ADC value of the central non-enhancing cavity, the centrally reduced diffusion sign, which is a possible indicator for TRA [13]. Another important limitation of the current study is the knowledge that TRA and TP often coexist within the same lesion, albeit in different ratios. Therefore, mean ADC values for the entire lesion could be confounded and, in turn, this could decrease the diagnostic accuracy of the ADC value in distinguishing TRA from TP. Nevertheless, when TP is present, this requires further treatment, regardless of the coexistence of TRA [16,37]. ADC values as an individual cut-off diagnostic, as well as radiologists’ readings of multiparametric data at certain timepoints, to distinguish TRA from TP are of limited value when compared to assessing temporal evolution. Hopefully, other metrics will be discovered that show to be more accurate in predicting TP from TRA at an earlier timepoint to tackle this diagnostic pitfall for better patient management. Including perfusion imaging in future studies could aid in distinguishing areas of recurrence from TRA. By figuring out a way to focus ADC analysis on areas of interest with elevated perfusion abnormalities, regions of recurrence could potentially be identified more readily.

This study does not take into account the methodological and technical variations of scanners, since all patients are from a single institute, yet did not perform as well as reported in recent meta-analyses. This could also be the reason for the notable difference in mean ADC values found compared to the meta-analysis [15]. Although the presented data are therefore homogeneous with regard to the acquisition protocol, this partially limits the translation of our results to other datasets, as the quantitative ADC cut-off values found by this study might differ from other groups. Therefore, the reported cut-off values cannot straightforwardly be generalized, and the need for universal standardization of MR protocols is highlighted once more.

## 5. Conclusions

This study shows significantly different mean ADC values between TP and TRA lesions on a group level. However, the use of ADC values as a diagnostic test to differentiate TP and TRA resulted in moderate diagnostic accuracy. This indicates that the use of semiquantitative assessment of the ADC maps in daily practice should only be used in repeated follow-ups with other imaging metrics in order to accurately distinguish TRA from TP.

## Figures and Tables

**Figure 1 cancers-15-04990-f001:**
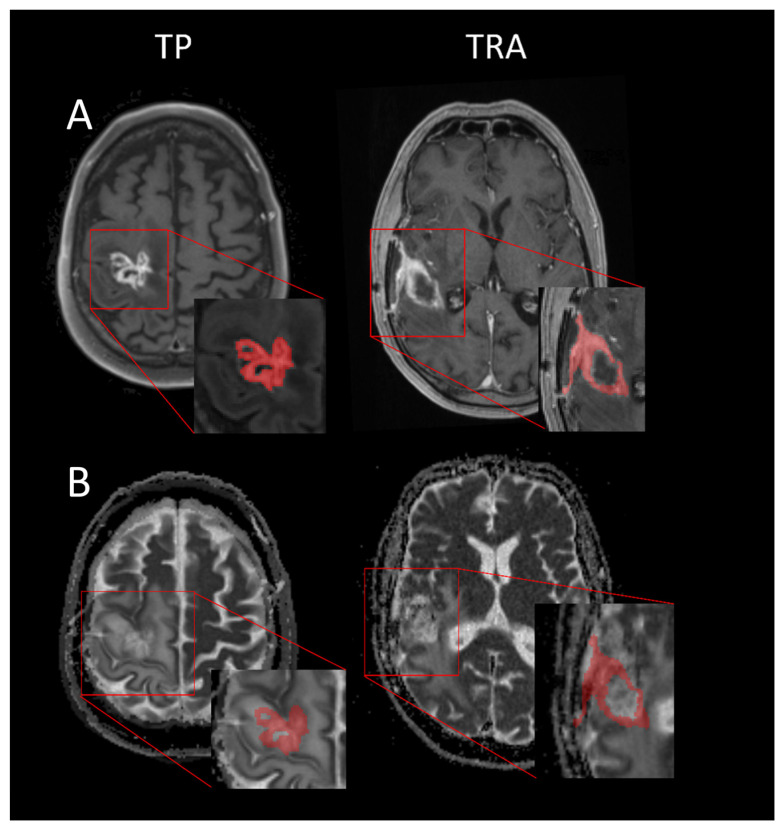
Axial T1-CE sequences and ADC maps of confirmed TP and TRA patients. (**A**) Contrast-enhanced T1-weighted images with segmenation; (**B**) ADC map with segmentation overlay The mean ADC values of the lesions in this figure are TP = 1.14 × 10^–3^ mm^2^/s, TRA = 1.68 × 10^–3^ mm^2^/s.

**Figure 2 cancers-15-04990-f002:**
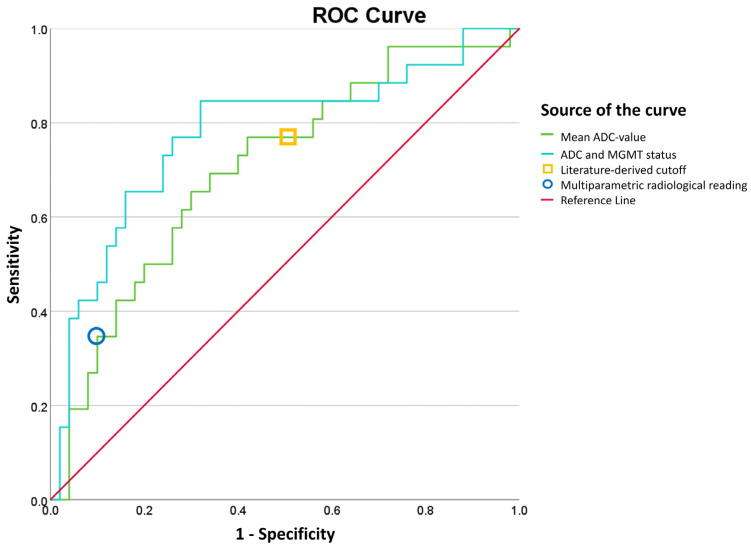
ROC analysis based on the ADC values found and literature cut-off. An AUC of 0.71 (95% confidence interval: 0.58–0.83) was yielded with regard to distinguishing TP and TRA on ADC values only. The AUC for combined MGMT status and mean ADC led to an AUC of 0.78 (95% CI: 0.66–0.90). The literature-derived cut-off ADC value of 1.33 × 10^–3^ mm^2^/s (13) yielded an AUROC of 0.635 (95% confidence interval: 0.505–0.764).

**Table 1 cancers-15-04990-t001:** Demographic data of the included patients and glioblastoma lesions as tumor progression (TP) or treatment-related abnormality (TRA).

Patient Info	TP (*n* = 50)	TRA (*n* = 26)
Gender	M = 26 F = 24	M = 17 F = 9
Mean age (years)	57 (SD 12)	64 (SD 11)
MGMT promotor methylation status	20 hypermethylated 22 non-hypermethylated	20 hypermethylated 3 non-hypermethylated
IDH status	44 wildtype 5 mutant	25 wildtype

**Table 2 cancers-15-04990-t002:** MRI examination protocol. (N/A: not applicable).

Parameter	T1-MPRAGE	T1-SPACE	T2	T2-FLAIR	DWI-Resolve	DSC-Perfusion
Repetition time (TR) (ms)	2100	600	5310	9000	4210	1350
Echo time (TE) (ms)	2.42	7.1	85	87	75	40
Inversion time (TI) (ms)	N/A	N/A	N/A	2500	N/A	N/A
B values	N/A	N/A	N/A	N/A	0, 1000	N/A
Slice thickness (mm)	1	1	5	5	5	5
Matrix size (Pixels)	256 × 256	256 × 256	256 × 256	256 × 256	192 × 192	128 × 128
Resolution (mm × mm)	1 × 1	1 × 1	1 × 1	1 × 1	1.33 × 1.33	2 × 2
Acquisition plane	sagittal	sagittal	transversal	transversal	transversal	Transversal
Acquisition time (min)	5:00	3:26	2:30	3:30	2:00	2:00

**Table 3 cancers-15-04990-t003:** Confusion matrix of primary diagnosis by radiologists based on 76 patients with a complete diagnosis.

Total = 76 Patients	Observed TP	Observed TRA	Total
Predicted TP	45	17	62
Predicted TRA	5	9	14
Total	50	26	76

## Data Availability

Data available on request due to restrictions eg privacy or ethical. The data presented in this study are available on request from the corresponding author. The data are not publicly available due to potential loss of anonymity from DICOM headers and scans.

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
