# Peer review of "Apparent Diffusion Coefficient Metrics to Differentiate between Treatment-Related Abnormalities and Tumor Progression in Post-Treatment Glioblastoma Patients: A Retrospective Study"

_cancers, 2023, doi:10.3390/cancers15204990_

Round 1
Reviewer 1 Report
The manuscript “Apparent diffusion coefficient metrics to differentiate between treatment related abnormalities and tumor progression in post-treatment glioblastoma patients: a retrospective study” represents an original study of glioblastoma, focusing on radiological (MRI-based) differential diagnosis between tumour recurrence versus treatment-induced changes in the tissues. Considering the dismal prognosis of glioblastoma, the topic undoubtedly is timely and important; the research on certain aspects of tumour diagnostics has high clinical relevance and adds continuous scientific evidence to the previous studies.
The contents of the manuscript correspond to the scope of the journal. The study design is sound; the results are detailed and well-illustrated.
There are only few suggestions to further improve the manuscript:
1) Authors have noted that MGMT promoter methylation status is significantly associated with the likelihood of tumour progression, compared to treatment-related abnormalities. At the same time, no significant differences in ADC values were shown between the groups showing promoter hypermethylation versus lacking it. Would it be possible to identify tumour progression more accurately (i.e., with better specifity and sensitivity), based on an algorithm, implementing both MGMT methylation status and ADV values?
2) In glioblastoma, several molecular subtypes have been identified both by gene expression and immunohistochemistry. These subtypes were shown to predict the efficacy of the treatment. See, please, Verhaak et al., 2010 (PMID: 20129251) and Jakovlevs et al., 2019 (PMID: 32146793) for gene expression-based and immunohistochemical subtyping, respectively. Molecular subtyping, esp. by immunohistochemistry, is easily available in diagnostic surgical pathology labs. Please, discuss if these molecular subtypes could serve as additional adjuncts to identify tumour progression, similarly to MDMT methylation status, along with MRI findings?
3) It would be reader-friendly to describe shortly the RANO criteria.
Finally, I would like to thank the authors for their work input and dedication. It was a pleasure and a true honour to review this manuscript.
The level of English language is generally good. Nevertheless, minor language editing might be useful to remove the few minor misprints (e.g., lines 18 and 31). Extra spaces seem to be present on lines 268, 270 and 345. Ref.18 ends with an unfinished phrase.
Author Response
Dear reviewer,
Thank you for your meticulous review of my paper. I do believe your suggestions have helped improve the contents of this manuscript.
Overview of alterations:
I have highlighted my changes. I have removed the double spaces further down the document, the lines have however shifted due to addition of extra text.
I have changed reference 18 to show the complete reference
I have added an analysis and included an updated ROC curve of MGMT status and ADC value in a logistic regression.
I have included proposed references and discussed molecular subtyping without going into too much detail for the sake of lenghtiness of manuscript.
I have shortly described RANO criteria.
As I said before, I do believe these changes are of benefit to the reader, with additional reading for those interested, more complete analysis and a better overview. Thank you once again for your clear and concise feedback with attention to detail.
I hope the new version is in line with what you envisioned during the feedback process.
Best regards,
Rik van den Elshout
Reviewer 2 Report
In this retrospective study, the authors looked at archived diffusion MRI data from glioblastoma (GBM) patients who were diagnosed post-treatment (i.e., maximal surgery followed by the Stupp protocol) with new contrast-enhancing lesions. Specifically, the authors analyzed one diffusion MRI metric—i.e., the apparent diffusion coefficient (ADC)—that is often employed in clinical practice to classify new contrast-enhancing lesions as either tumor progression (TP) or treatment-related abnormalities (TPA). The accuracy of this discrimination is critically important in the clinical management of post-treatment GBM patients. To better understand the clinical utility of this MRI metric, the authors sought to validate in their analysis a certain ADC cutoff value previously reported in recent meta-analyses for its discriminative properties between TP and TPA.
The present study confirms that mean ADC values differ between TP and TPA, with the latter showing significantly higher ADC values than the former. However, the study also finds these ADC values to be less discriminative than expected as they show significantly more overlapping compared to what was found by previous studies. From the analysis of their data, the authors conclude that the sole use of ADC cutoffs in isolation (i.e., at only one timepoint) is not feasible for making the accurate diagnosis of TP versus TPA during the radiological follow-up of post-treatment GBM patients. Therefore, the authors highlight the importance of multiparametric radiologic follow-up in these patients. I found the study to be well-designed and well-executed and with convincing conclusions. Importantly, the authors also include a comprehensive discussion on the strengths and limitations of their study to the great benefit of the potential reader.
Author Response
Dear reviewer,
Thank you for your generous feedback on my manuscript, I am happy to see it lives up to your standards.
I have had some suggestions for improvement from the other reviewer, please see the highlights in the new document. I hope these changes are in line with your already positive review.
Among the changes I have included:
Additional analysis of ADC value combined with MGMT with binary logistic regression, including a new ROC figure, discussion topic on molecular subtyping and explained the RANO criteria.
Thank you for your time and meticulous reading of my manuscript.
Best regards,
Rik van den Elshout